# LLM-based Validated Code Translation and Repair for High-Performance Software

## Abstract

We present an LLM-based code translation and repair framework called TRI-anslate, which translates existing code written in an arbitrary source language to an arbitrary target language and validates that the output code adheres to desired properties via testing. Existing work has shown that LLMs are remarkable at code translation and repair tasks. Furthermore, specialized fine-tuned or distilled LLMs can extend these capabilities to handle niche languages, perform syntax repair with relatively small cost, or perform semantic repair taking into account common errors. However, the most robust currently available tools that leverage these LLMs assign all these distinct subtasks to a single LLM with a feedback loop from a validation tool. Further, they rely on a rigid set of possible errors as part of the corrective feedback from the validator or verifier. By contrast, TRI-anslate allows for a user-specified error set and leverages 3 separate LLM feedback loops to fully utilize the capability of LLMs specialized for generation, syntactic repair, and semantic repair. This also avoids wasting context of later LLMs on the correction conversation of previous LLMs. We conduct an extensive evaluation, showcasing the advantage of TRI-anslate over the existing work using the same setup ($\approx 8\%$ increase comparing the base model, $\approx 45\%$ for the fine-tuned model in CUDA to OpenMP Target Offloading Translation). We also demonstrate how being able to choose different models per subtask allows TRI-anslate to outperform LASSI using any of the individual models, and highlight the extensibility of TRI-anslate by documenting the effort required to add a new translation task (CUDA to SYCL).

## 1 Introduction

Porting legacy and machine-specific code, via automated translation techniques, to more modern forms that are compatible and performant on the latest machines is increasingly important, especially in High-Performance Computing (HPC) settings Świniarski & Derezińska (2025); Bandarupalli (2025); Ranasinghe et al. (2025); Diggs et al. (2024); Godoy et al. (2023); Smith & Garcia (2020); Gel et al. (2017). The motivations for automated code translation are many, including improved security, translating institutional knowledge preserved in legacy code into a modern form, and lowering maintenance cost by leveraging modern tooling and software infrastructure Dearing et al. (2024); Juckeland et al. (2017). Further, automated code translation tools can be cheaper and less error-prone. By contrast, manual translation is costly, time-consuming, and error-prone, leading to interest in automated code translation tools Pennycook et al. (2019); Juckeland et al. (2017); Du et al. (2012); Trott et al. (2012); Noaje et al. (2016); Cooper et al. (2003).

This interest has prompted significant industrial research and development in transpilers , rule-based code translation tools which map program syntax from a given input programming language into the target while preserving semantics of the programs being translated. Developing such tools require one to have expertise in both (the source and target) languages, and further require accurate modeling of the semantics of each language for validation. Another important requirement is that the generated code in the target language must be human readable, for otherwise the entire point of porting legacy code to modern infrastructure is rendered moot. These requirements make development of transpilers expensive to develop in general, unapproachable for niche languages, and can introduce subtle errors unless the tool is designed and developed extremely carefully in order to properly handle semantic edge cases.

In recent years, prompted by the rise of Large Language Models (LLMs), their efficacy in handling many code tasks, cost-effectiveness, flexibility, and adaptability, we have witnessed groundbreaking developments in LLM-base code translation tools . As long as one has sufficient high-quality training data on the source and target languages, LLMs have shown great success in generating translated code which has all the right properties, i.e., is syntactically correct, seems to preserve program semantics, and even appears to be human-written. LLMs have also shown success in feedback-based code repair, for certain classes of syntactic and semantic errors Chen et al. (2021); Roziere et al. (2023); Joshi et al. (2023), via a natural pairing of an LLM and a validation tool. The LLM provides generations or repair attempts, and a validation tool judges the attempt and provides corrective feedback Song et al. (2025); Guo et al. (2023).

However, these tools do have some weaknesses, such as their inability to scale to large programs as well as traditional transpilers Li et al. (2022); Austin et al. (2021). In addition, general foundation models perform poorly in languages with low data or opaque feedback from verification tools Lu et al. (2021). The second weakness is somewhat overcome by fine-tuning models with specialized datasets. However, fine-tuning typically results in a loss of the generality, i.e., the tool becomes excellent in the narrow context for which it was fine-tuned, but then loses other capabilities that it previously possessed prior to fine-tuning Hu et al. (2022); Dettmers et al. (2023). This results in LLMs which can address one component of the translation task, but performs poorly for the others.

## 1.1 OUR APPROACH

To address the need for better LLM-based code translation tools, we introduce TRI-anslate, a fully automated end-to-end code translation tool that leverages three LLMs in separate feedback re-prompting loops for generation, syntax repair, and semantic repair, where the syntatic and semantic repair LLMs in turn get feedback from a validator (tester). This system of dividing the translation into these three subtask loops allows the models to not inherit unrelated context from the past, and further be more specialized through different system prompts. Also, when particular subtasks require larger models and more effort, this system allows for bulking up a certain stage without being forced to use an expensive model on the entire process. Similarly, if a smaller model is capable of handling a specific stage, this system allows for greater efficiency. We demonstrate the capability and flexibility of TRI-anslate through comparison against an existing LLM-based HPC code translation tool, evaluating the success rate on an existing translation task, and showcasing the ease of implementing an arbitrary new translation task.

## 1.2 CONTRIBUTIONS

In more detail, we make the following contributions:

1. **Novel code translation method via 3 LLMs looped with in-context verifier feedback**:
   We present an LLM-based code translation and repair tool called TRI-anslate, which leverages 3 separate LLM and validator feedback loops to ensure code adheres to specification of both syntax and semantics. TRI-anslate has several advantages over prior LLM-base code translation tools. It is more *flexible* in that the the various components of the tool are plug-and-play and one can leverage fine-tuned LLMs designed for different translation subtasks. TRI-anslate is also more scalable and efficient because the design of the triple LLM corrective feedback loop enables the context windows to be used more efficiently. Further, we have been careful in making implementation of TRI-anslate to be agnostic to the source and target languages. With appropriate fine-tuning, validators, and prompt specification, TRI-anslate has the ability to handle any pair of source and target languages.

2. **Extensive experimental evaluation against SOTA**: Using a selection of diverse scientific computing kernels collected in the HeCBench benchmark suite Jin & Vetter (2023), we compare TRI-anslate against LASSI Dearing et al. (2024), a state-of-the-art code translation tool for scientific software, to showcase the value of the multi-feedback loop design. TRI-anslate successfully translated 5/62 ($\approx 8\%$) more kernels on the base model and 28/62 ($\approx 45\%$) on a fine-tuned model. We show that by choosing unique LLMs for the different feedback loops, we can outperform all options for LASSI using the same models. We also showcase the ease of extending TRI-anslate for arbitrary new translation tasks, highlighting its flexibility.

## 2 RELATED WORK

**Transpilers.** These tools have decades of rich history, which we won't be able to do justice to here. We briefly mention some recent transpilers such as `java2python` Melhase et al. (2016) and `py2java` Fomin (2019), and TSS code converter TSS (2023), a commercial J2P transpiler. Despite decades of work, it is well-known that human-written transpilers are error-prone and are expensive to write and maintain as they require expertise in many small details of both the source and target languages. This also results in unavailability when either language is without a large following or high demand.

By contrast, LLM-based code translation tools, including ours, are cheaper to maintain and develop, since it is data-driven. The primary challenge is in finding or creating sufficiently large high-quality dataset of equivalent source-target program pairs. Such tools are also more flexible than human-written transpilers, i.e., can be easily adapted to newer settings without requiring significant effort.

**Transformer and ML-based Code Translation Tools.** Early ML approaches included encoder-based CodeBERT Feng et al. (2020), decoder-based CodeGPT Lu et al. (2021) and CodeT5 Wang et al. (2021), an encoder-decoder system that uses developer supplied identifiers to aid translation. PLBART Ahmad et al. (2021) is a separate unified transformer model trained via denoising autoencoding for a range of NL and Programming Language (PL) tasks.

**LLM-based methods.** Recent trends in the area have been gravitating towards using LLMs for translation paired with the use of validation or verification tools during fine-tuning as well as inference. The use of validation and/or verification tools enable one to be certain that the translated code adheres to desired properties (most importantly, the semantic input/output equivalence between the source and target programs). Examples include TransCoder-ST Roziere et al. (2022) an unsupervised framework for code translation that employs self-training through automated unit tests to assess equivalence between source and target code implementations. Wang et al. (2022) uses reinforcement learning and pass/fail compiler feedback to fine-tune for code generation. PPOCoder Shojaee et al. (2023), adds CodeBLEU-inspired reward signals during fine-tuning. The Cotran tool Jana et al. (2024) uses a variant of Reinforcement Learning with Symbolic Feedback (RLSF) Jha et al. (2024) in order to fine-tune a pair of LLMs back-to-back with the goal of translating Python to Java and back.

In the High-Performance computing domain, LASSI uses compiler and test case feedback during translation inference to check the LLM generated solutions are valid. Our approach extends these LLM-based translation methods by subdividing inference into three subtasks (namely, generation, syntactic repair, semantic repair), allowing for the usage of specialized LLMs for each subtask.

## 3 TRI-ANSLATE DESIGN AND ARCHITECTURE

### 3.1 OVERVIEW

TRI-anslate is designed to fully utilize the advancements in LLM-based translation subtasks while retaining validation of results and overall translation success. The infrastructure is also easily extensible, with easy to edit prompts, LLMs, and validation tools. TRI-anslate also maintains useful statistics and runtime information for the user.

Figure 1 shows an overview of the infrastructure for the experiments of this paper. Note that all components within the figure are able to be customized or swapped out for existing tools to fit a desired arbitrary translation task. TRI-anslate handles all the data management and bookkeeping and edges of the figure, and gives interfaces to adjust the nodes. The structure consists of:

1. **Preprocessor** - prepares the input for translation
2. **Generation Loop** - performs initial translation attempts and filters trivial mistakes
3. **Syntactic Repair Loop** - validates the syntax of the generation and attempts repairs
4. **Semantic Repair Loop** - validates the semantics of the generated code and attempts repairs

Should any of these components fail to succeed beyond a set number of times per component (modifiable heuristics), the process will restart so as to avoid getting stuck too long on a single attempt.

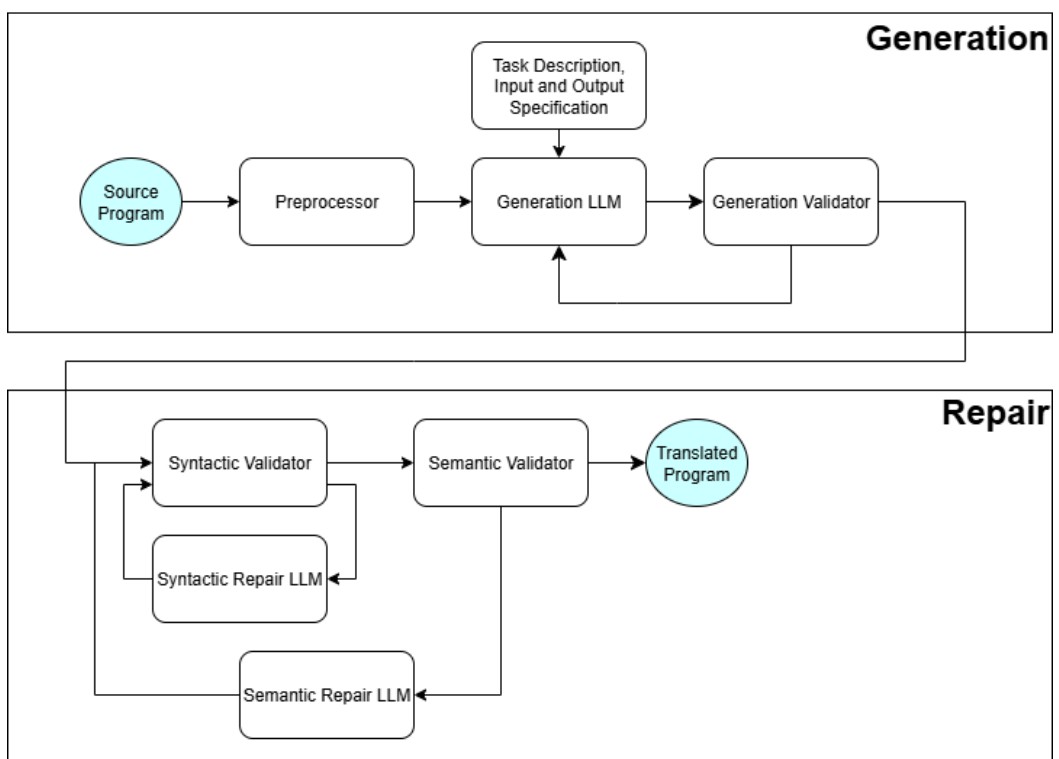

Figure 1: TRI-anslate Infrastructure Overview

The infrastructure was implemented in python, providing modifiable callback-style functions and prompt dictionary, allowing full user control over the translation while providing the infrastructure to abstract away the data management and other underlying annoyances.

## 3.2 PREPROCESSOR

Prior to translation, we first put the input code through a preprocessor. This module allows the input to be transformed into a form more amenable to LLM input or more relevant to the task at hand.

For our experiments, this component served two functions. One goal was extracting the individual kernel functions from the input because we are performing function by function translation, and secondly, we add in a hint about the interface we want it to adhere to so individual functions are compatible. These inputs came from HeCBench Jin & Vetter (2023), a collection of heterogeneous computing benchmarks written with CUDA, HIP, SYCL/DPC++, and OpenMP-4.5 target offloading for studying performance, portability, and productivity. The benchmark suite is also utilized by LASSI for evaluation.

## 3.3 GENERATION LOOP

The next component of TRI-anslate is the generation loop. This loop is responsible for performing the initial translation attempt and checking that it obeys the requirements of the problem. The LLM is provided the system prompt and initial generation prompt (see Appendix D for the exact prompts used during experiments), and a generation is made. The validator confirms that the generation adheres to the requirements and returns feedback and a status that matches up directly with a prompt in the prompt dictionary if it does not. This feedback and status are used in reprompting based on the specific issue discovered. Once the validator has certified that the generation passes requirements, it allows the current attempt to break out of this loop.

For our experiments, the requirements we checked are that the codeblock existed in the attempt, and that it contained either OpenMP Target offloading statements or SYCL code depending on the experiment being run. This helps to rule out cases where the LLM generates no code at all, wasting repair time, or sequential code, which can cause false positives. Note that a similar check that the code contains the goal programming model was not present in the LASSI tool, but we added it for fair comparison during experiments.

## 3.4 SYNTACTIC REPAIR LOOP

The syntactic repair loop is responsible for getting the code syntactically correct, which in most cases means able to compile. Similar to before, the LLM is given the system prompt, but unlike before, we check the correctness before prompting. If the validator detects any issues, it matches these up to the reprompting options and uses the feedback and prompt to fix the mistakes in the code.

For our experiments, this validator replaced the function in an existing correct translation in HeCBench with the LLM-generated code and attempted to compile. The compiler errors were fed directly back into a repair prompt. Our method does allow for different prompts of fine granularity for each of the possible compiler error types, but we opted instead to use a general prompt and the provided output from the compiler as the feedback as this exact validation was also done in LASSI.

## 3.5 SEMANTIC REPAIR LOOP

Lastly, we have a semantic repair loop. This component generally uses errors in runtime behavior of the code or formal semantic checks to guide repair. Firstly, we have the validator perform checks on the code and use feedback and prompt dictionary matching as before. However, note that after performing semantic repair, we need to ensure the LLM did not modify the code into an attempt which no longer compiles, therefore each attempt links back to another pass through the syntactic repair loop.

In our experiments, the semantic checks were tests already present within the HeCBench files we substituted into. Therefore, we ran the tests and gave failures as feedback. This is definitely an informal guarantee of translation success, however it matches the approach used by LASSI, and could be substituted by something more rigorous depending on the task.

## 4 EXPERIMENTAL EVALUATION

We aim to answer two research questions by benchmarking and analyzing the results of TRI-anslate:

- **RQ1 (Comparison Against SOTA):** How does TRI-anslate compare against SOTA LLM-based code translation tools?
- **RQ2 (Design Flexibility):** How adaptable is TRI-anslate to new code translation tasks?

## 4.1 EVALUATION SETUP

To collect the results referenced in this section, we run on a machine running Ubuntu (version 22.04), an AMD EPYC 9454 (Genoa) system with two 48-core CPUs, 1.5 TB RAM, and an NVIDIA H100 NVL GPU using Ollama (version 0.11.10) application programming interface through Python (version 3.10.12) to run the models.

The kernels from HeCBench are specifically chosen from diverse scientific domains for a better representation of real world problems.

The lables in Table 1 are used to refer to LLM triples used in TRI-anslate (B.#) and individual LLMs used in LASSI (C.#) during the experiments. Also, ChatPort-32B is a version of qwen2.5-coder:32b-instruct fine-tuned using translation examples of either CUDA to OpenMP Offloading or CUDA to SYCL depending on the task being tested. This fine-tuning was done in accordance with the method described in Pophale et al. (2025).

Exact model versions, seeds, temperature settings, prompts, and verbose breakdowns of the benchmark results are available in the appendicies B, C, D, and E.

Table 1: TRI-anslate and LASSI LLMs

| ID | Generation LLM | Syntactic Repair LLM | Semantic Repair LLM |
|---|---|---|---|
| B.1 | qwen2.5-coder:32b-instruct | qwen2.5-coder:32b-instruct | qwen2.5-coder:32b-instruct |
| B.2 | ChatPort-32B | ChatPort-32B | ChatPort-32B |
| B.3 | qwen2.5-coder:32b-instruct | qwen2.5-coder:7b-instruct | qwen2.5-coder:32b-instruct |
| B.4 | ChatPort-32B | qwen2.5-coder:7b-instruct | qwen2.5-coder:32b-instruct |
| B.5 | ChatPort-32B | qwen2.5-coder:7b-instruct | gpt-oss:20B |
| C.1 | qwen2.5-coder:32b-instruct | | |
| C.2 | ChatPort-32B | | |
| C.3 | qwen2.5-coder:7b-instruct | | |

Table 2: CUDA to OpenMP Results

| LLM Triple | Solved / 62 |
|---|---|
| TRI-anslate Single Code LLM B.1 | 32 |
| TRI-anslate Fine-Tuned LLM B.2 | 45 |
| TRI-anslate Smaller Syntactic Repair B.3 | 50 |
| TRI-anslate Fine-Tuned Gen. & Smaller Syntax Repair B.4 | 47 |
| TRI-anslate F-T Gen. & Smaller Syntax & Chain-Of-Thought Semantic Repair B.5 | 44 |
| LASSI w/ qwen2.5-coder:32b-instruct C.1 | 27 |
| LASSI w/ ChatPort-32B C.2 | 17 |
| LASSI w/ qwen2.5-coder:7b-instruct C.3 | 31 |

## 4.2 RQ1: COMPARISON AGAINST SOTA

In order to evaluate our tool against the current state-of-the-art, we test CUDA to OpenMP translation against LASSI. Edits were made to LASSI in order for fair comparison. These edits are adding a timelimit cutoff of 10 minutes per testcase, adding an additional validation check that the code generated by the LLM uses the desired model (OpenMP Target Offloading), and adjusted the prompts and compilation to be for function-to-function translation instead of whole program, including adding the desired interface of the translated function and a validation check that the generated code matches. These same requirements were imposed on TRI-anslate, using the same feedback in prompts in event of a mismatched function interface or the code did not use the desired programming model.

The results of running TRI-anslate and LASSI are in table 2. Some interesting observations come from these tests.

Firstly, both LASSI and TRI-anslate benefited from switching to the smaller code model. Intuition predicted that the smaller model would suffer a performance hit if used for the entirety of translation, but would have less of an issue if utilized only for the syntactic repair subtask of the translation. The fact that LASSI also outperformed using the smaller model compared to the larger version of the same model was unexpected. It is difficult to pinpoint an exact reason for this. One potential reason is the speed of generation allowed both tools to generate more attempts within the timelimit. Another interesting result was that the fine-tuned LLM outperformed the basemodel when used for all three stages, succeeding on 45 cases instead of 32, but when comparing the basemodel with smaller syntax repair against the fine-tuned model with smaller syntax repair, the fine-tuned model translated 3 fewer tests successfully. This implies that the generation of the basemodel is potentially stronger than the fine-tuned LLM, but the fine-tuned LLM is better at the repair stages, the opposite of what the fine-tuning was expected to produce.

B.1 and C.1, along with B.2 and C.2 show apples-to-apples comparison where LASSI and TRI-anslate are using the exact same model. One apparent result from these experiments is that using 3 feedback loops allowed us to achieve better results than LASSI even using the exact same model and setup. TRI-anslate succeeded on $5/62 \, (\approx 8\%)$ more test cases for using qwen2.5-coder:32b-instruct, and $28/62 \, (\approx 45\%)$ more test cases when using the fine-tuned ChatPort-32B model. Although there

Table 3: CUDA to SYCL TRI-anslate Results

| LLM Triple | Solved / 62 |
|---|---|
| Single Code LLM B.1 | 46 |
| Fine-Tuned LLM B.2 | 38 |

is no way to be certain, the reasoning we feel is most likely for this occurrence is the ability to have the chat reset and distinct system prompts per subtask. This clears out the context used by prior stages of translation and guides the current step better, allowing the model to best match the subtask at hand without overloading the model context with unrelated fixes made previously. This seems to affect the fine-tuned model much more significantly, perhaps due to the loss of generality from fine-tuning affecting the model's capability to switch subtasks.

Additionally, the comparison between B.3 and C.3 shows that while TRI-anslate was able to benefit from the smaller model during syntactic repair only, LASSI cannot choose to use different models mid-translation. In both the cases of LASSI only using the small model and when only using the larger model, it performed worse and could not reach the 50 solved by the combination of the two used by TRI-anslate.

Another result is that switching from the single code LLM of qwen2.5-coder:32b-instruct to the fine-tuned ChatPort-32B LLM saw an increased success in translation where LASSI actually saw a decrease in successful translations. This fine-tuned model may be overloaded when the context is kept and there is no system prompt to change the focus of the model to the new subtask.

## 4.3 RQ2: DESIGN FLEXIBILITY

In order to showcase the flexibility of our tool, we implemented a different translation task, CUDA to SYCL, on the same test set. SYCL is a very different programming model than OpenMP, and equally or more niche compared to the target offloading semantics of OpenMP. This task also does not have any available traditional transpilers, and there are not that many resources in terms of datasets and code examples. This makes CUDA to SYCL translation another perfect task for LLMs, more specifically fine-tuning LLMs to best address the problem.

In order to extend our tool to handle this new translation task, we have to edit a few components of the process. The prompt dictionary file is changed to have the prompts in D.2 instead. The generation validator is changed to ensure SYCL was generated instead of OpenMP Target. The syntactic validator calls a SYCL compiler instead of OpenMP and replaces the function in a SYCL oracle instead of an OpenMP oracle. No changes were required for the preprocessor because we are taking from the same dataset, and no changes were required for the semantic validator to run the testcases.

In general, changing to a different translation task involves swapping out the preprocessor, validators, and prompts. The prompts are in a simple json file and the preprocessor and validators are all done through python callback-style functions which provide all the relevant information regarding the state of the translation and a scratchpad to store data for other validator calls later in the process. No part of the system is language specific, allowing for fair results regardless of the source and target language. In Table 3, we see that out of the same tests, TRI-anslate was able to adapt and find decent success on translation to a completely different target programming model.

TRI-anslate is already prepared to run a user provided list of files/folders and create detailed formatted reports of the results. This allows for testing and comparing different model triples on many different benchmarks easily. TRI-anslate also has built in intranode parallelism support, allowing for all available GPUs to run models attacking the current testcase.

## 5 DISCUSSION

### 5.1 LIMITATIONS

One limitation of the work is that we evaluate TRI-anslate on only a single benchmark set for these code translations. The HeCBench benchmark suite is the source for all the test kernels we translate, the source for the fine-tuning of the ChatPort LLM (different examples than the test set), and the source of the test cases for evaluation of the translations. This is unfortunately mostly unavoidable, as HeCBench is the only suite of programs for high-performance code with valid translated implementations in multiple different niche HPC programming models.

Another limitation is the work only compares to LASSI and not a large suite of existing code translation tools including traditional transpilers. Part of the value of the work of LLM-based code translation is the flexibility to handle more niche source and target languages via fine-tuning, which we tested with our experiments. In these cases, there are no existing transpilers and it would be prohibitively difficult to write one. Also, many existing LLM-based approaches do not have validation or prompting easily mapped to this task.

Although our method can be extended to do more rigorous checking, one possible limitation for our evaluation is that we can have false positives. Possible examples include using compiler-specific code which would not be portable, test set benchmarks with weak tests, or some sort of cheating where the model hardcodes in the answers to tests that it failed. This concern is valid, however the experiments were done under the same models and same validation criteria, so the tools are completely equally evaluated. In the future, adding multiple compilers to test, or generating tests randomly and getting expected output via the oracle implementation may be a quick way to attempt to make the validation more robust. Also, formal verifiers exist as an option if the code is pre-annotated with invariants or proof goals.

### 5.2 FUTURE WORK

Currently, the TRI-anslate infrastructure is capable of parallelizing across multiple GPUs running models for a given test case. This parallelism assists the scalability of the approach, but has not been thoroughly examined nor evaluated. Also, this parallelizes multiple chat threads for the same test case for more attempts, although it may be more practically useful to instead parallelize across different test cases to leverage all the computing resources for efficient translation of large scale programs.

We also would like to investigate more thoroughly the capabilities of models to handle the generation and repair. Specifically, we believe that agentic-style or commercial models could potentially prove better at some subtasks. More testing needs to be done to find the specific triple of model types that balance results and efficiency.

Also, our infrastructure naturally creates good training data for fine-tuning repair models. We save all attempts, including those before and after syntactic and semantic success with the validator feedback. This is an opportunity to extend the framework for a symbiotic relationship with existing work on fine-tuning for repair.

## 6 CONCLUSIONS

We have presented TRI-anslate, a novel framework for performing automatic code translation through the use of a triple validated LLM feedback loop system. TRI-anslate allows for separation of subtasks, specialization of models, and the refreshing of model context, leading to TRI-anslate outperforming LASSI. TRI-anslate is able to successfully perform more translations in a one-to-one comparison, using the same models ($\approx 8\%$ for base model, $\approx 45\%$ for fine-tuned), and with the ability to use specific models with specific subtasks, surpass any possible LASSI run using the same models. TRI-anslate is also language agnostic and shown to be easy to extend for use in new arbitrary code translation tasks, able to handle CUDA to SYCL translation without issue.

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

## A  USAGE OF LLMs IN PAPER PREPARATION

LLMs were used to assist the writing of this paper. They were used to:

- Aid or polish writing
- Retrieve related work
- Write scripts and functions for analyzing/summarizing experimental results
- Assist in table and figure formatting
- Creating and cleaning bibliography

More specifically: Some sentences were given to an LLM to be rewritten for clarity. Many python scripts to load in the data and generate LaTeX tables were created via refining the initial attempt by an LLM. Some formatting such as the indentation of LLM triple information in B was suggested by an LLM. The field of LLM-based software engineering is rapidly developing, so LLMs were used to retrieve related work to ensure the already cited work in the area was sufficient.

## B  TRI-ANSLATE LLM INFO

### B.1  LLM TRIPLE #1 : SINGLE CODE LLM

GENERATION

> **Model:** `qwen2.5-coder:32b-instruct`
> **Temperature:** 0.2
> **Seed:** 82

SYNTACTIC REPAIR

> **Model:** `qwen2.5-coder:32b-instruct`
> **Temperature:** 0.2
> **Seed:** 82

SEMANTIC REPAIR

> **Model:** `qwen2.5-coder:32b-instruct`
> **Temperature:** 0.2
> **Seed:** 82

### B.2  LLM TRIPLE #2 : FINE-TUNED LLM

GENERATION

> **Model:** `ChatPort-32B`
> **Base Model:** `qwen2.5-coder:32b-instruct`
> **Temperature:** 0.2
> **Seed:** 82

SYNTACTIC REPAIR

> **Model:** `ChatPort-32B`
> **Base Model:** `qwen2.5-coder:32b-instruct`
> **Temperature:** 0.2
> **Seed:** 82

SEMANTIC REPAIR

> **Model:** `ChatPort-32B`
> **Base Model:** `qwen2.5-coder:32b-instruct`
> **Temperature:** 0.2
> **Seed:** 82

## B.3  LLM TRIPLE #3 : SMALLER SYNTACTIC REPAIR

GENERATION

> **Model:** `qwen2.5-coder:32b-instruct`
> **Temperature:** 0.2
> **Seed:** 82

SYNTACTIC REPAIR

> **Model:** `qwen2.5-coder:7b-instruct`
> **Temperature:** 0.2
> **Seed:** 82

SEMANTIC REPAIR

> **Model:** `qwen2.5-coder:32b-instruct`
> **Temperature:** 0.2
> **Seed:** 82

## B.4  LLM TRIPLE #4 : FINE-TUNED GENERATION & SMALLER SYNTACTIC REPAIR

GENERATION

> **Model:** `ChatPort-32B`
> **Base Model:** `qwen2.5-coder:32b-instruct`
> **Temperature:** 0.2
> **Seed:** 82

SYNTACTIC REPAIR

> **Model:** `qwen2.5-coder:7b-instruct`
> **Temperature:** 0.2
> **Seed:** 82

SEMANTIC REPAIR

> **Model:** `qwen2.5-coder:32b-instruct`
> **Temperature:** 0.2
> **Seed:** 82

## B.5 LLM TRIPLE #5 : FINE-TUNED GENERATION & SMALLER SYNTACTIC REPAIR & CHAIN-OF-THOUGHT SEMANTIC REPAIR

GENERATION

> **Model:** `ChatPort-32B`
> **Base Model:** `qwen2.5-coder:32b-instruct`
> **Temperature:** 0.2
> **Seed:** 82

SYNTACTIC REPAIR

> **Model:** `qwen2.5-coder:7b-instruct`
> **Temperature:** 0.2
> **Seed:** 82

SEMANTIC REPAIR

> **Model:** `gpt-oss:20b`
> **Temperature:** 0.2
> **Seed:** 82

# C LASSI LLM INFO

## C.1 MODEL #1

> **Model:** `qwen2.5-coder:32b-instruct`
> **Temperature:** 0.2
> **Seed:** 82

## C.2 MODEL #2

> **Model:** `ChatPort-32B`
> **Base Model:** `qwen2.5-coder:32b-instruct`
> **Temperature:** 0.2
> **Seed:** 82

## C.3 MODEL #3

> **Model:** `qwen2.5-coder:7b-instruct`
> **Temperature:** 0.2
> **Seed:** 82

# D PROMPTS

The following variables in prompts are replaced by data mid-translation:

**+SRC CODE+**      The original code to be translated

**+GENERATION+**      The current attempt generated by the LLM

**+FEEDBACK+**      The feedback from the last validation check (could be compiler, tests, etc.)

## D.1 CUDA TO OPENMP PROMPTS

GENERATION

| Prompt Type | Prompt |
|---|---|
| System | You are a programming model translation expert which will translate functions written in CUDA C++ to equivalent functions using OpenMP Target Offloading. For every piece of code, produce: 1. A translated code block surrounded with "' that preserves the semantics and parallelism. 2. A description section explaining your thought process for the translation. Rigorous checks will determine if your translation attempt is successful. |
| Initial Generation | Please perform translation of the following CUDA code to OpenMP Target Offloading maintaining the OpenMP function interface: "'+SRC CODE+"' |
| Missing Codeblock | Your generation attempt was missing the codeblock surrounded by "'. Please retry generation. Here is the original code to translate to OpenMP Target Offloading: "'+SRC CODE+"' |
| No OpenMP Target Offloading | Your generation attempt did not use the Target Offloading capabilities of OpenMP. Please retry generation. Here is the original code to translate to OpenMP Target Offloading: "'+SRC CODE+"' |

SYNTACTIC REPAIR

| Prompt Type | Prompt |
|---|---|
| System | You are a code syntax repair expert. You will be provided the code and the error message. Use the error information to repair the code and fix the error. |
| Compiler Error | The following OpenMP Target Offloading code is uncompilable. Please repair the code to compile correctly. Code: +GENERATION+ Compiler Output: +FEEDBACK+ |

SEMANTIC REPAIR

| Prompt Type | Prompt |
|---|---|
| System | You are an expert at fixing logical errors in code. You will be provided the original code written in CUDA C++ and the attempt at recreating the code using OpenMP Target Offloading. Identify any logical issues and correct the OpenMP Target Offloading code. |
| Test Error | The following OpenMP Target Offloading code fails tests that ensure that the behavior of the code is equivalent to a correct implementation written in CUDA C++. Original Code: +SRC CODE+ OpenMP Code: +GENERATION+ Feedback: +FEEDBACK+ |

## D.2 CUDA TO SYCL PROMPTS

GENERATION

| Prompt Type | Prompt |
|---|---|
| System | You are a programming model translation expert which will translate functions written in CUDA C++ to equivalent functions using SYCL. For every piece of code, produce: 1. A translated code block surrounded with "' that preserves the semantics and parallelism. 2. A description section explaining your thought process for the translation. Rigorous checks will determine if your translation attempt is successful. |
| Initial Generation | Please perform translation of the following CUDA code to SYCL maintaining the SYCL function interface: "'+SRC CODE+"' |
| Missing Codeblock | Your generation attempt was missing the codeblock surrounded by "'. Please retry generation. Here is the original code to translate to SYCL: "'+SRC CODE+"' |
| No SYCL Kernel | Your generation attempt did not use the SYCL capabilities. Please retry generation. Here is the original code to translate to SYCL: "'+SRC CODE+"' |

SYNTACTIC REPAIR

| Prompt Type | Prompt |
|---|---|
| System | You are a code syntax repair expert. You will be provided the code and the error message. Use the error information to repair the code and fix the error. |
| Compiler Error | The following SYCL code is uncompilable. Please repair the code to compile correctly. Code: +GENERATION+ Compiler Output: +FEEDBACK+ |

SEMANTIC REPAIR

| Prompt Type | Prompt |
|---|---|
| System | You are an expert at fixing logical errors in code. You will be provided the original code written in CUDA C++ and the attempt at recreating the code using SYCL. Identify any logical issues and correct the SYCL code. |
| Test Error | The following SYCL code fails tests that ensure that the behavior of the code is equivalent to a correct implementation written in CUDA C++. Original Code: +SRC CODE+ SYCL Code: +GENERATION+ Feedback: +FEEDBACK+ |

# E  VERBOSE BENCHMARK RESULTS

## E.1  CUDA TO OPENMP

| Test | B.1 | B.2 | B.3 | B.4 | B.5 | C.1 | C.2 | C.3 |
|---|---|---|---|---|---|---|---|---|
| ace::boundaryConditionsPhi | ✓ | ✓ | ✓ | ✓ | ✓ | ✓ | ✓ | ✓ |
| ace::swapGrid | ✓ | ✗ | ✓ | ✓ | ✓ | ✓ | ✓ | ✓ |
| adam::adam | ✓ | ✓ | ✓ | ✓ | ✓ | ✗ | ✗ | ✗ |

| | | | | | | | | |
|---|---|---|---|---|---|---|---|---|
| aidw::AIDW_Kernel | ✗ | ✓ | ✓ | ✓ | ✓ | ✓ | ✗ | ✓ |
| aop::generate_paths_kernel | ✓ | ✗ | ✗ | ✗ | ✗ | ✓ | ✗ | ✗ |
| aop::update_cashflow_kernel | ✗ | ✗ | ✗ | ✗ | ✗ | ✓ | ✗ | ✗ |
| assert::perfKernel | ✓ | ✓ | ✓ | ✓ | ✓ | ✓ | ✗ | ✓ |
| atomicCost::woAtomicOnGlobalMem | ✓ | ✓ | ✓ | ✓ | ✓ | ✗ | ✗ | ✗ |
| atomicPerf::SingleRangeAtomicOnGlobalMem | ✓ | ✓ | ✓ | ✓ | ✓ | ✗ | ✗ | ✗ |
| attention::attention_kernel1 | ✓ | ✓ | ✓ | ✓ | ✓ | ✓ | ✓ | ✓ |
| attention::attention_kernel2 | ✓ | ✓ | ✓ | ✓ | ✓ | ✗ | ✓ | ✓ |
| attention::attention_kernel3 | ✓ | ✓ | ✓ | ✓ | ✓ | ✗ | ✓ | ✓ |
| axhelm::axhelm_n3 | ✗ | ✗ | ✗ | ✗ | ✗ | ✗ | ✗ | ✗ |
| background-subtract::findMovingPixels | ✗ | ✓ | ✗ | ✓ | ✓ | ✓ | ✗ | ✓ |
| background-subtract::merge | ✗ | ✓ | ✓ | ✓ | ✓ | ✓ | ✗ | ✓ |
| background-subtract::updateBackground | ✗ | ✓ | ✓ | ✓ | ✓ | ✓ | ✗ | ✓ |
| background-subtract::updateThreshold | ✗ | ✓ | ✓ | ✓ | ✓ | ✓ | ✗ | ✓ |
| backprop::kernel_adjust_weights | ✗ | ✓ | ✓ | ✓ | ✓ | ✓ | ✗ | ✓ |
| burger::core | ✗ | ✓ | ✓ | ✓ | ✓ | ✓ | ✗ | ✗ |
| bwt::reconstruct_sequence | ✓ | ✓ | ✓ | ✓ | ✓ | ✗ | ✗ | ✗ |
| channelShuffle::ChannelShuffleNCHWKernel | ✗ | ✗ | ✓ | ✗ | ✗ | ✗ | ✗ | ✗ |
| channelSum::ChannelSumNCHW | ✗ | ✗ | ✓ | ✗ | ✗ | ✗ | ✗ | ✗ |
| channelSum::ChannelSumNHWC | ✗ | ✗ | ✗ | ✗ | ✗ | ✗ | ✗ | ✗ |
| chemv::chemv_kernel0 | ✗ | ✓ | ✗ | ✓ | ✓ | ✗ | ✗ | ✗ |
| chi2::chi_kernel | ✗ | ✓ | ✗ | ✓ | ✓ | ✗ | ✓ | ✗ |
| clink::lstm_inference | ✗ | ✓ | ✓ | ✓ | ✓ | ✓ | ✓ | ✗ |
| cmp::compute_semblances | ✓ | ✓ | ✓ | ✓ | ✓ | ✓ | ✗ | ✓ |
| cmp::init_c | ✗ | ✓ | ✓ | ✓ | ✓ | ✗ | ✗ | ✓ |
| cmp::init_half | ✗ | ✗ | ✓ | ✓ | ✓ | ✓ | ✗ | ✓ |
| concat::concat | ✓ | ✗ | ✓ | ✗ | ✗ | ✗ | ✗ | ✗ |
| convolution3D::conv3d_s3 | ✓ | ✗ | ✓ | ✗ | ✗ | ✗ | ✗ | ✗ |
| cross::cross3_kernel | ✓ | ✗ | ✓ | ✗ | ✗ | ✗ | ✗ | ✗ |
| dense-embedding::dense_esuhm | ✓ | ✗ | ✓ | ✗ | ✗ | ✗ | ✗ | ✗ |
| fdtd3d::finite_difference | ✓ | ✓ | ✓ | ✓ | ✓ | ✗ | ✗ | ✗ |
| flip::flip_kernel | ✓ | ✓ | ✓ | ✓ | ✓ | ✗ | ✗ | ✗ |
| gd::compute | ✓ | ✓ | ✓ | ✓ | ✓ | ✓ | ✗ | ✓ |
| glu::glu_kernel | ✓ | ✓ | ✓ | ✓ | ✓ | ✗ | ✓ | ✓ |
| goulash::gate | ✗ | ✓ | ✓ | ✓ | ✓ | ✗ | ✗ | ✓ |
| haccmk::haccmk_kernel | ✗ | ✓ | ✓ | ✓ | ✓ | ✗ | ✗ | ✗ |
| heat::initial_value | ✓ | ✓ | ✓ | ✓ | ✓ | ✓ | ✓ | ✓ |
| hotspot3D::hotspot3d | ✗ | ✓ | ✗ | ✓ | ✓ | ✗ | ✗ | ✓ |
| hwt1d::dwtHaar1D | ✓ | ✓ | ✓ | ✓ | ✗ | ✓ | ✓ | ✓ |
| ising::update_lattice | ✓ | ✓ | ✓ | ✓ | ✓ | ✗ | ✗ | ✓ |
| iso2dfd::iso_2dfd_kernel | ✓ | ✓ | ✓ | ✓ | ✓ | ✓ | ✓ | ✓ |
| laplace::red_kernel | ✗ | ✗ | ✓ | ✓ | ✓ | ✓ | ✓ | ✓ |
| lif::lif | ✓ | ✓ | ✓ | ✓ | ✓ | ✓ | ✓ | ✗ |
| mcpr::compute_probs_unitStrides | ✗ | ✓ | ✓ | ✓ | ✓ | ✓ | ✗ | ✓ |
| mrc::MRCGradient2 | ✗ | ✓ | ✓ | ✓ | ✓ | ✓ | ✗ | ✓ |
| nbody::accelerate_particles | ✗ | ✓ | ✓ | ✓ | ✓ | ✓ | ✓ | ✓ |
| nlll::nll_loss_forward_reduce2d_kernel | ✗ | ✗ | ✗ | ✓ | ✗ | ✗ | ✗ | ✗ |
| overlay::DetectionOverlayBox | ✗ | ✗ | ✓ | ✗ | ✗ | ✗ | ✗ | ✗ |
| p4::postprocess | ✓ | ✓ | ✗ | ✗ | ✗ | ✗ | ✗ | ✗ |
| page-rank::reduce | ✗ | ✗ | ✗ | ✗ | ✗ | ✗ | ✗ | ✗ |
| particle-diffusion::Simulation | ✗ | ✓ | ✓ | ✓ | ✓ | ✓ | ✗ | ✓ |
| permute::permute_kernel | ✗ | ✓ | ✓ | ✓ | ✗ | ✓ | ✗ | ✗ |
| projectile::CalculateRange | ✓ | ✓ | ✓ | ✓ | ✓ | ✗ | ✗ | ✗ |
| softmax::softMax | ✓ | ✓ | ✓ | ✓ | ✓ | ✗ | ✓ | ✗ |
| swish::SwishGradientKernel | ✓ | ✓ | ✓ | ✗ | ✗ | ✗ | ✗ | ✗ |
| tqs::TaskQueue_gpu | ✓ | ✓ | ✓ | ✓ | ✓ | ✗ | ✓ | ✓ |

| vanGenuchten::vanGenuchten | ✓ | ✓ | ✓ | ✓ | ✓ | ✗ | ✓ | ✗ |
|---|---|---|---|---|---|---|---|---|
| vol2col::vol2col_kernel | ✗ | ✗ | ✓ | ✗ | ✗ | ✗ | ✗ | ✗ |
| winograd::winograd_conv2d | ✓ | ✓ | ✗ | ✓ | ✓ | ✗ | ✗ | ✓ |

## E.2 CUDA TO SYCL

| Test | B.1 | B.2 |
|---|---|---|
| ace::boundaryConditionsPhi | ✓ | ✓ |
| ace::swapGrid | ✓ | ✓ |
| adam::adam | ✓ | ✗ |
| aidw::AIDW_Kernel | ✓ | ✓ |
| aop::generate_paths_kernel | ✗ | ✗ |
| aop::update_cashflow_kernel | ✗ | ✗ |
| assert::perfKernel | ✗ | ✗ |
| atomicCost::woAtomicOnGlobalMem | ✗ | ✗ |
| atomicPerf::SingleRangeAtomicOnGlobalMem | ✗ | ✗ |
| attention::attention_kernel1 | ✓ | ✓ |
| attention::attention_kernel2 | ✓ | ✓ |
| attention::attention_kernel3 | ✓ | ✓ |
| axhelm::axhelm_n3 | ✗ | ✗ |
| background-subtract::findMovingPixels | ✗ | ✗ |
| background-subtract::merge | ✓ | ✓ |
| background-subtract::updateBackground | ✓ | ✓ |
| background-subtract::updateThreshold | ✗ | ✓ |
| backprop::kernel_adjust_weights | ✓ | ✓ |
| burger::core | ✓ | ✓ |
| bwt::reconstruct_sequence | ✓ | ✓ |
| channelShuffle::ChannelShuffleNCHWKernel | ✗ | ✗ |
| channelSum::ChannelSumNCHW | ✓ | ✗ |
| channelSum::ChannelSumNHWC | ✓ | ✗ |
| chemv::chemv_kernel0 | ✗ | ✗ |
| chi2::chi_kernel | ✓ | ✓ |
| clink::lstm_inference | ✗ | ✗ |
| cmp::compute_semblances | ✓ | ✓ |
| cmp::init_c | ✓ | ✓ |
| cmp::init_half | ✓ | ✓ |
| concat::concat | ✓ | ✗ |
| convolution3D::conv3d_s3 | ✓ | ✗ |
| cross::cross3_kernel | ✓ | ✗ |
| dense-embedding::dense_esuhm | ✓ | ✗ |
| fdtd3d::finite_difference | ✗ | ✗ |
| flip::flip_kernel | ✓ | ✓ |
| gd::compute | ✓ | ✓ |
| glu::glu_kernel | ✓ | ✓ |
| goulash::gate | ✓ | ✓ |
| haccmk::haccmk_kernel | ✓ | ✓ |
| heat::initial_value | ✓ | ✓ |
| hotspot3D::hotspot3d | ✓ | ✓ |
| hwt1d::dwtHaar1D | ✓ | ✓ |
| ising::update_lattice | ✓ | ✓ |
| iso2dfd::iso_2dfd_kernel | ✓ | ✓ |
| laplace::red_kernel | ✓ | ✓ |
| lif::lif | ✓ | ✓ |

| | | |
|---|:---:|:---:|
| mcpr::compute_probs_unitStrides | ✓ | ✓ |
| mrc::MRCGradient2 | ✓ | ✓ |
| nbody::accelerate_particles | ✓ | ✗ |
| nlll::nll_loss_forward_reduce2d_kernel | ✗ | ✓ |
| overlay::DetectionOverlayBox | ✓ | ✗ |
| p4::postprocess | ✓ | ✓ |
| page-rank::reduce | ✓ | ✓ |
| particle-diffusion::Simulation | ✓ | ✓ |
| permute::permute_kernel | ✓ | ✓ |
| projectile::CalculateRange | ✓ | ✗ |
| softmax::softMax | ✓ | ✓ |
| swish::SwishGradientKernel | ✗ | ✗ |
| tqs::TaskQueue_gpu | ✗ | ✓ |
| vanGenuchten::vanGenuchten | ✓ | ✓ |
| vol2col::vol2col_kernel | ✓ | ✗ |
| winograd::winograd_conv2d | ✗ | ✗ |

