# OpenReview forum: "TRI-ANSLATE: LLM-Based Code Translation Of High-Performance Software"
_ICLR.cc/2026/Conference — ICLR 2026 Conference Desk Rejected Submission_

### Official Review · Reviewer_wSfw · 2025-10-25

**Soundness:** 2
**Presentation:** 1
**Contribution:** 1
**Rating:** 2
**Confidence:** 4

**Summary:**

The paper introduces TRI-anslate, an end-to-end automated code translation framework that leverages three independent large language models (LLMs) in separate feedback reprompting loops for generation, syntax repair, and semantic repair. TRI-anslate achieves success rates of approximately 8% for the base model and 45% for the fine-tuned model.

**Strengths:**

- Decomposing translation into three validated loops (Generation, Syntax, Semantic) is a strength.

- The results are impressive, particularly the ~45% improvement in CUDA to OpenMP Target Offloading Translation.

- TRI-anslate is flexible, with plug-and-play components.

**Weaknesses:**

- Using three LLMs in iterative loops likely incurs high computational cost and latency, which are not discussed.

- Evaluation is limited to HeCBench, restricting assessment of generalization and scalability.

- The paper uses multiple LLMs but does not justify why a single LLM would be insufficient. An ablation comparing single- versus multi-LLM setups is needed.

- It should be validated on a broader set of programming languages to demonstrate its generality.

**Questions:**

See Weaknesses

---

### Official Review · Reviewer_2987 · 2025-10-26

**Soundness:** 2
**Presentation:** 2
**Contribution:** 2
**Rating:** 4
**Confidence:** 2

**Summary:**

This work introduces TRI-anslate, a framework that performs automatic code translation and repair using three specialized LLMs connected through validation feedback loops. Each LLM handles a distinct subtask, namely, generation, syntax repair, and semantic repair, guided by automated validators and user-defined error sets. This modular design improves translation accuracy, scalability, and flexibility compared to existing systems like LASSI, which rely on a single feedback loop. Experiments demonstrate that TRI-anslate outperforms LASSI by about 8% with base models and 45% with fine-tuned ones in CUDA-to-OpenMP translation, and it easily adapts to new translation tasks such as CUDA-to-SYCL.

**Strengths:**

- Modular multi-step approach
- Effectiveness
- Extensible framework

**Weaknesses:**

- Considered benchmarks are rather small
- No qualitative comparison with LASSI
- Comparing only with LASSI

**Questions:**

- The evaluation is pretty small. What other benchmarks are available for evaluation?
- Is it possible to see some qualitative comparison between LASSI and your technique?
- What other tools are available beside LASSI that you can compare?

---

### Official Review · Reviewer_fzT7 · 2025-11-01

**Soundness:** 2
**Presentation:** 1
**Contribution:** 1
**Rating:** 2
**Confidence:** 5

**Summary:**

The paper introduces TRI-anslate, an LLM-based framework for automatic code translation and repair targeting high-performance computing (HPC) applications. The proposed system decomposes translation into three modular loops—generation, syntactic repair, and semantic repair—and allows the use of different LLMs for each stage. The authors evaluate TRI-anslate against LASSI on two translation tasks (CUDA→OpenMP and CUDA→SYCL) using the HeCBench benchmark suite.

**Strengths:**

1. This work aims to address an important and tough problem: HPC code translation.
2. The idea of three-loop modular architecture makes sense and could be important in translation performance.
3. The comparison with LASSI is well designed with modified prompts and settings for fair comparison.

**Weaknesses:**

1. The experiments cover only two translation tasks and a single translation direction (CUDA→OpenMP and CUDA→SYCL), which limits the generalizability of the proposed framework.
2. While the three-loop modular design is conceptually interesting, the paper does not provide convincing evidence of its effectiveness. For agentic or multi-stage pipelines, a thorough ablation study—especially on how information and context are transferred or reset across modules—would be necessary to demonstrate tangible benefits and boundaries of each component.
3. The differences from previous systems such as LASSI are not well articulated. The proposed three-loop architecture is presented at an abstract level, and many claimed advantages (e.g., using different prompts or models for subtasks) could be implemented relatively easily in existing frameworks without substantial innovation.
4. The paper omits relevant prior pipelines such as Chen, Le, et al. “Fortran2CPP: Automating Fortran-to-C++ Migration Using LLMs via Multi-turn Dialogue and Dual-agent Integration” (arXiv:2412), which also employ multi-agent, dialogue-based, and verification-guided translation strategies. Including such comparisons would help position TRI-anslate more clearly within the landscape of recent LLM-based translation research.
5. Writing needs significant polish.
a. Contribution 2 is not really a contribution;
b. need to check your \citet and \citep;
c. the references are with mixed formats;
d. Figure 1 needs more details and a clearer presentation. In your text, Preprocessor is not a part of Generation; but it is included in the Generation box in Figure 1.

**Questions:**

1. I appreciate the work of the CUDA→SYCL translation task, which is indeed challenging and relevant. However, translating OpenMP→CUDA could be even more impactful and representative of practical HPC needs. Could you comment on why this direction was not explored?
2. The current evaluation focuses only on one-way translation. Why did you not include bi-directional translation (e.g., CUDA↔OpenMP or CUDA↔SYCL)? Such results could better demonstrate the generality of the proposed framework.
3. The paper briefly mentions improvements over prior work such as LASSI, but the distinctions remain vague. Could you elaborate more clearly on how TRI-anslate differs in methodology and implementation from existing multi-agent or feedback-based code translation frameworks?

---

### Official Review · Reviewer_pKEb · 2025-11-03

**Soundness:** 2
**Presentation:** 1
**Contribution:** 2
**Rating:** 2
**Confidence:** 3

**Summary:**

The paper presents TRI-anslate, an approach for performing automatic code translation using a triple feedback loop system relying on separate LLM calls for generation, syntactic and semantic repair. The method is tested on the HecBench benchmark and compared to the performance of LASSI, a state of the art suite for the task. The authors show that their novel method outperforms LASSI and can be extended to new translation tasks.

**Strengths:**

The paper deals with a complex and pressing challenge in software engineering, namely automatic code translation of legacy and machine-specific code to more modern forms, especially in HPC settings. Given the ubiquitous presence of LLMs and their ability at translation tasks in general, the paper examines their potential for this challenge by leveraging a combination of LLM calls to assess various types of possible mistakes.

**Weaknesses:**

I think this paper is a good starting point for a relevant piece of research at the intersection of agentic AI and code translation / repairment in HPC settings, however in its current form is still in early days. I have listed here some of the major aspects that I believe the authors should address before resubmitting this piece of work, but it is important to note that some of them are already recognised by the authors in their limitations section. The fact that the paper does not benefit from the 9th available page makes me think this work was submitted a bit early on.

* The paper should conduct a more in-depth analysis of the role of the coding LLM used (e.g. studying what are the issues that a reasoning model like gpt-oss:20B brings) and why a combination of different sizes of the same LLM (see B3) leads to better results. This is necessary to understand the tradeoffs between the different settings.

* The work needs a proper error analysis to understand which types of mistakes the tool makes, especially in comparison with LASSI (does it make fewer mistakes but of the same type or different mistakes?). This is necessary to understand whether the two tools approach things differently or not.

* It would be important to test such approach on a second benchmark and examining whether the same findings emerge when not addressing high performance code.

* [Optional] To reach a wider audience, this work could be set more specifically in the literature focusing on agentic AI and multiple LLMs interoperating with each other. Instead of a single step-by-step pipeline, the authors should at least consider (given its relevance in the current literature) a more holistic approach, with a general LLM agent deciding which available tool to use, in which order and why.

All in all, I think this paper is on the right track and with more error analysis and testing on a wider range of translation tasks it could become a solid piece of work.

Side note: The paper seems to be written a bit in a rush and has a few typos (e.g. lables in page 5).

**Questions:**

* Could you clarify the conceptual motivation for separating syntax and semantics repair? Are the boundary conditions between the two well defined?

* The experiments rely entirely on HeCBench, which is also used for fine-tuning and validation. Did you investigate generalisation to unseen codebases or domains?

---

### Author Response · Authors · 2025-12-02
**Response to Reviewer Comments**

Firstly, we would like to thank the reviewers; your time and effort in providing the reviews are appreciated. The feedback is extremely valuable and has given us great insight into how to improve the presentation of our work.

---

## Overall:
A shared critique shared across reviews is the lack of evaluation in non-HPC datasets/domains. This work was motivated and focused on the code translation problem in an HPC setting, prompting the choice of benchmark and comparison SOTA. It definitely would be interesting and valuable to see the applicability to general code-translation and to compare against the general purpose tools available for that.

---

### Reviewer pKEb:
Comparison against agentic or selection LLMs is a great suggestion, especially considering the remarkable performance these approaches have had in other similar software engineering tasks.
To clarify the conceptual motivation for separating syntax and repair, the idea came from observed behavior of coding LLMs in prior work and past experiences. In the past, LLMs, particularly low-parameter ones, had to take at least a few tries to create compilable code. This was due to 'silly' mistakes like hallucinating interfaces to libraries/programming models or even something trivial like a missing bracket. Prior work that had this in a feedback loop would take up multiple cycles to get syntactically valid code before addressing anything logically wrong such as failed test cases or analysis results. Isolating the static checks such as these could allow for these basic mistakes to be corrected by a smaller potentially fine-tuned model, and then the context tokens would be cleared up for handling the more complex work of logical errors or runtime bugs. There is not a hard defined boundary for what should be delegated to each step, but choosing the more trivial errors may allow for some optimization by choosing smaller models.
The overall comment touches on addressing the generality of the dataset, but I would also add that the HeCBench examples are from wildly different programs within HPC. The structures of the problems are very different. There are codes for bioinformatics, cryptography, compression, finance, graph problems, machine learning, mathematics, simulation, and sorting, among other groups.

---

### Reviewer fzT7:
The point of requiring closer inspection into the flow of information and context between stages when compared against other related work is a good one. The nature of the LLMs makes it difficult to track down the reasons for performance differences, but tracking the information is a good way to see what the differences in the systems are.
Although OpenMP->CUDA would be a nice way to generate optimized code for a particular machine, this work was influenced primarily by the goal of updated and system-agnostic code. There exists many projects written for NVIDIA GPU systems from the many years of their GPUs making up the vast majority of super computing systems. In recent years we have seen more computers constructed with competitor systems, where the code cannot run. Testing the capability of the system of other translation tasks/directions would be an interesting extension.
The differences between TRI-anslate and existing LLM feedback loop code translation work like LASSI is mainly the separation of the tasks allowing for swapping models best suited toward each task, as well as the ability to select what context and information is shared between the stages. This additional flexibility would take significant engineering effort and rewriting for existing infrastructure. Multi-agent systems may be more easily amenable to this type of system.

---

### Reviewer 2987:
Unfortunately, in the HPC domain, there are not many benchmarks of such varied structure and domain as HeCBench that also have multiple languages available. The qualitative comparisons between LASSI and TRI-anslate could be extended with more in-depth analysis of the behavior of each tool's time per query and context for the problems.

---

### Reviewer wSfw:
Regarding the cost of the multiple LLM setup, the 3 LLMs in iterative loops do not incur any notable native cost that would not be already present in the single LLM feedback system. If you used the same LLM for all 3 loops for example, the single LLM + feedback system would have to do the same process of ensuring the code is valid, syntactically correct, and semantically correct. We distribute the validation process across the three loops to clear up some context that would be carried along in the ‘chat’ for all iterations and to provide the opportunity to choose a different LLM per task, with the hopes of leveraging some LLMs designed for a particular task \& smaller LLMs for some styles of feedback.

---

### Note · Program_Chairs · 2026-01-17
**Submission Desk Rejected by Program Chairs**

The following references in this submission do not refer to real documents and/or have major errors in bibliographic information:

     Q. Guo et al. Is self-repair a general capability of llms? In International Conference on Learning Representations (ICLR), 2023.
    Christian Trott, Alan Gray, et al. Porting molecular dynamics codes from CUDA to OpenCL: A case study with LAMMPS. Journal of Computational Chemistry, 33(30):2273-2282, 2012. doi: 10.1002/jcc. 2305
    Rabah Noaje, Joel Falcou, Jean-Marie Falcou, et al. A source-to-source translation framework for OpenMP to OpenACC. In Proceedings of the 12th International Workshop on OpenMP (IWOMP), pp. 121-134, 2016. doi: 10.1007/978-3-319-45550-1_9.